# Developmental exposure to the brominated flame retardant DE-71 reduces serum thyroid hormones in rats without hypothalamic-pituitary-thyroid axis activation or neurobehavioral changes in offspring

**Louise Ramhøj**[1]*, **Terje Svingen**[1], **Karen Mandrup**[1], **Ulla Hass**[1], **Søren Peter Lund**[2], **Anne Marie Vinggaard**[1], **Karin Sørig Hougaard**[2,3], **Marta Axelstad**[1]

**1** National Food Institute, Technical University of Denmark, Kgs. Lyngby, Denmark, **2** National Research Centre for the Working Environment, Copenhagen, Denmark, **3** Department of Public Health, University of Copenhagen, Copenhagen, Denmark

* louram@food.dtu.dk

## Abstract

Polybrominated diphenyl ethers (PBDEs) are legacy flame retardants for which human exposure remains ubiquitous. This is of concern since these chemicals can perturb development and cause adverse health effects. For instance, DE-71, a technical mixture of PBDEs, can induce liver toxicity as well as reproductive and developmental toxicity. DE-71 can also disrupt the thyroid hormone (TH) system which may induce developmental neurotoxicity indirectly. However, in developmental toxicity studies, it remains unclear how DE-71 exposure affects the offspring's thyroid hormone system and if this dose-dependently relates to neurodevelopmental effects. To address this, we performed a rat toxicity study by exposing pregnant dams to DE-71 at 0, 40 or 60 mg/kg/day during perinatal development from gestational day 7 to postnatal day 16. We assessed the TH system in both dams and their offspring, as well as potential hearing and neurodevelopmental effects in prepubertal and adult offspring. DE-71 significantly reduced serum T4 and T3 levels in both dams and offspring without a concomitant upregulation of TSH, thus inducing a hypothyroxinemia-like effect. No discernible effects were observed on the offspring's brain function when assessed in motor activity boxes and in the Morris water maze, or on offspring hearing function. Our results, together with a thorough review of the literature, suggest that DE-71 does not elicit a clear dose-dependent relationship between low serum thyroxine (T4) and effects on the rat brain in standard behavioral assays. However, low serum TH levels are in themselves believed to be detrimental to human brain development, thus we propose that we lack assays to identify developmental neurotoxicity caused by chemicals disrupting the TH system through various mechanisms.

**Data Availability Statement:** All relevant data are within the manuscript and its Supporting Information file.

**Funding:** This study was funded by the Danish Environmental Protection Agency, Ministry of Environment and Food of Denmark. KS Hougaard received support from FFIKA, Focused Research Effort on Chemicals in the Working Environment, from the Danish Government. The funders had no role in study design, data collection and analysis, decision to publish, or preparation of the manuscript.

**Competing interests:** The authors have declared that no competing interests exist.

## Introduction

Disrupted thyroid hormone (TH) signaling during perinatal life can adversely affect the developing brain. This is because proper brain development is critically dependent on spatiotemporally controlled thyroid hormone action [1, 2]. The TH system, however, comprise an intricate network of interconnected organs and signaling events. Thus, TH system disruption by chemical substances can occur by different mechanisms. This includes inhibition of thyroperoxidase (TPO), the sodium/iodide symporter (NIS), deiodinases, dehalogenases, and transporters such as the monocarboxylate transporter 8 (MCT8). It can also occur by displacement of TH from the serum distributor protein transthyretin (TTR) or augmentation of liver metabolism and excretion of TH through induction of metabolizing enzymes and excretion-transporters [1, 3, 4].

Exposure to compounds targeting any of the abovementioned TH system components can have different effects on the TH system itself, as well as on the developing brain. One mechanism and effect pattern that is well described in rats is TPO inhibition (reducing TH synthesis) leading to activation of the hypothalamic-pituitary-thyroid axis (HPT-axis). In this scenario, there is a dose-dependent correlation between reduced serum T4 concentrations and altered brain development manifesting with both morphological and behavioral effects [5–12]. However, for some TH system disrupting chemicals not acting through this TPO-mediated mechanism, it is unclear if there is the same correlation between low serum T4 concentrations and TH-mediated brain effects. This includes environmental chemicals that presumably disrupt the TH system by either binding to TTR or induce metabolizing enzymes in the liver, for instance triclosan [13, 14], polychlorinated biphenyls (PCBs) [15], the perfluorinated compound perfluorohexane sulfonate (PFHxS) [14, 16] and potentially also the polybrominated diphenyl ether (PBDE) mixture DE-71 (PBDE) [17–19].

Now banned, PBDEs were for decades used as flame retardants in both industrial and consumer products. Today, human exposure continues due to environmental contamination, persistence and bioaccumulation. Thus, the primary exposure routes are food and house dust [20]. DE-71 is a technical mixture of PBDEs and its congeners are still found in human serum, breast milk and house dust [20, 21]. In humans and animals, exposure to PBDEs has been associated with a range of effects including neurotoxicity, disruption of the reproductive and thyroid hormone systems.

In adult rats DE-71 markedly reduces serum thyroxine (T4) concentrations (>70% reduction), while TSH levels remain unaffected, or slightly increased [17, 22–24]. A similar effect pattern is seen in perinatally exposed rat offspring, where DE-71 reduces serum T4 by more than 70% without inducing TSH levels to any great degree [17, 18, 24]. One question is if such marked reductions in T4, without a concomitant compensatory increase in TSH, can perturb neurodevelopment and cause cognitive or other toxicological effects in exposed offspring similarly to those caused by TPO-inhibitors. Based on a small number of developmental exposure studies with DE-71, it may be assumed that it does not. However, no definite conclusions can be drawn from the available data.

Previous toxicity studies with perinatal DE-71 exposure have reported marked reduction in T4 (70–90% in high exposure groups), yet without consistent adverse behavioral effects in the offspring [17–19]. In some of the studies, however, group sizes were relatively small which could impair their ability to detect subtle effects. In addition, the studies performed cognitive testing at different developmental ages using slightly different methodologies. Exposure windows also varied slightly between studies, as did the rat strains used; Long-Evans [17], Sprague-Dawley [18] or Wistar [19]. Finally, the potential effects on HPT-axis regulation remains poorly examined. Three studies report on TSH levels in DE-71 exposed pups; one with a 30%

increase [18] and two others with no changes to TSH [17, 24], and there are no thorough histological examinations of pup thyroid glands. Together, this makes it difficult to conclude on overall effect patterns on the TH system after perinatal exposure to DE-71.

To enable a more informed conclusion about any potential adverse effects on behavior following developmental exposure to DE-71, as well as address other unclear relationships as outlined above, we performed a robust *in vivo* toxicity study. The study design and endpoints are aligned with our previous studies of TPO inhibitors propylthiouracil, methimazole and amitrole [6, 10] (refer to Table 1 for summary results).This was done to answer the outstanding question whether there is a direct correlation between low serum T4 and neurobehavioral effects of DE-71, since only few and sporadic effects have been found in other studies investigating neurodevelopmental effects [17–19]. Furthermore, we wished to clarify if DE-71 can activate the HPT-axis in a manner consistent with TPO inhibition and whether it is plausible that DE-71 causes dose-dependent and correlated effects on serum T4 and TH-mediated brain development.

## Materials and methods

### Chemicals

The test compound was DE-71 (a technical mixture of penta-brominated diphenyl ethers (BDE), lot 7550OK20A), a commercial mixture of PBDEs (kind gift from Dr. Kevin Crofton at the U.S. Environmental Protection Agency). A gas chromatography coupled with high-resolution mass spectrometry (GC-HRMS) analysis of the congener content in our DE-71 solutions showed an approximate distribution of congeners in this lot of DE-71 as follows: ~29% tetra-PBDE (BDE-47), 59% pentaPBDE (of which ~50% BDE-99 and ~9% is BDE-100) and ~9% hexaPBDE (of which ~5% BDE-153 and ~4% is BDE-154). Corn oil (Sigma-Aldrich, St. Louis, MO, USA) was used as control compound and vehicle for all treatments.

### Animals and treatment

Two animal studies were conducted as depicted in Fig 1, using 40 and 66 time-mated pregnant Wistar rats (HanTac: WH, Taconic Europe, Ejby, Denmark) that were received on gestation day (GD) 3 of pregnancy (day of plug-detection designated GD1) and pseudo-randomly divided into groups. The expected day of delivery, GD23, was designated PD1 irrespective of actual day of delivery. A detailed description of experimental setup, reproductive toxicity outcomes, postnatal growth and disruption of the reproductive hormone systems have been reported previously [25]. In short, we found no evidence for overt toxicity in dams or pups, but shorter anogenital distance and reduced prostate weight in male offspring, as well as decreased mammary gland outgrowth in both sexes [25].

Study 1 comprised 4 treatment groups of 10 dams each, exposed to vehicle control (corn oil), or DE-71 at 20, 40 or 60 mg/kg bodyweight (bw)/day from GD7 to PD14, except the day of delivery. Study 2 was conducted in two balanced blocks and comprised three treatment groups of 22 dams each, exposed to vehicle control (corn oil), or DE-71 at 40 or 60 mg/kg bw/day from GD7 to PD16, except day of delivery. All exposure were administered by oral gavage at a constant volume of 2 ml/kg bw/day. Dams were housed pairwise in semitransparent plastic cages (15 × 27 × 43 cm) until GD17 and individually thereafter. Conditions were environmentally controlled and with reverse light/dark cycles (light from 9 pm-9 am). The animals were provided standard Altromin 1314 diet *ad libitum* (soy and alfalfa-free, Altromin GmbH, Lage, Germany). The iodine and selenium contents were 1.52 mg/kg and 0.26 mg/kg, respectively. Acidified tap water was provided *ad libitum* in PSU bottles (84-ACBTO702SU, Tecniplast, Buguggiate, Italy).

**Table 1. Developmental DE-71 exposure studies of the thyroid hormone system and brain development in rats.**

| Exposure to DE-71 | TH/TSH | Thyroid gland | Developmental neurotoxicity and hearing | Reference |
|---|---|---|---|---|
| *Studies including only thyroid hormone system endpoints* | | | | |
| 0, 0.1, 1, 10 or 30 mg/kg/day to SD dams GD6-PND21 (wafer, oral) (n = 6–9) | Dam PD21 tT4 and fT4↓ up to ~50%, tT3↔, TSH↔ Pup PD21 tT4↓ up to 85%, fT4↓ up to 75%, tT3↔, TSH↔ | - | - | [34] (Bansal et al., 2014) |
| 0 or 18 mg/kg/day to SD dams GD6-LD18 (intragastric) (n = 3–7) | Dam LD19 tT4↓45%, tT3↔, TSH↔ Pup PND18 tT4↓75%, tT3↔, TSH↔ | Dam thyroid gland weights were obtained but results not stated | - | [24] (Ellis-Hutchings et al., 2006) |
| 0, 1.7, 10.2 or 30.6 mg/kg/day to LE dams from GD6-PND21 (oral gavage) (n = 7–9) | Male pup: PND4 tT4↓ up to 50%, tT3↔ PND21 tT4↓ up to 75%, tT3↔ | - | - | [4] (Szabo et al., 2009) |
| 0, 1, 10 or 30 mg/kg/day to LE dams GD6-PND21 (oral gavage) (n = unclear, maybe 38–48 dams per group for both GD20 and PND22) | Dam GD20: tT4↓ 48% in high dose, tT3↔ Dam PND22: tT4↓45%, tT3↔ Fetal GD20: tT4↓ (more than 15%, control group very close to LOQ) Pup PND4: tT4↓ up to 40%, tT3↔ Pup PND14: tT4↓ up to 66%, tT3↔ | - | - | [35] (Zhou et al., 2002) |
| *Studies also including developmental neurotoxicity endpoints* | | | | |
| 0, 0.3, 3.0 or 30 mg/kg/day to SD dams GD1-PND 21 (cookie, oral) (n = 20–22 litters, TH: n = 9–10, TSH: n = 20–22, thyroid histo n = 8–10) | Dam PND21: tT4 and tT3 ↓ (data not shown) Male pup PND21: tT4↓ up to ~95%, tT3 ↓ up to 40% Female pup PND21: tT4↓ up to ~90%, tT3↓ up to 40% Pup PND21, TSH↑ ~30% in high dose | Male pup PND21: increased epithelial cell height in high dose | Brain weight PND21, 50, 105, 250: ↔ except PND21 raw brain weight ↓ in low dose and relative brain weight ↑ in high dose PND21 (both sexes), Motor activity PND 16, 55, 110 and 230: ↔, except small changes in rearing PND110 Acoustic startle PND20 and 90: potentiation on PND90 indicating delayed effects on sensory reactivity Beam test PND 33, 60: ↔ Emergency latency PND35, 80: ↔ Morris Water Maze PND 235: ↔ Nicotine-induced activity PND450: ↔ | [18] (Bowers et al., 2015) |
| 0 or 30 mg/kg/day to Wistar rat pups PND5-22 (oral gavage) (Open field: n = 17, radial maze: n = 11, T4: n = 10) | Male pup PND 23: tT4↓ ~50% Female pup PND 23: tT4↓ ~50% | Male pup PND23: thyroid gland weight↔ Female pup PND23: thyroid gland weight↔ | Brain weight PND23↔ Open field PND42 and 70: ↔ motor activity both sexes RAM PND100: reference memory deficit, females only | [19] (De-Miranda et al., 2016) |
| 0, 1.7, 20.2 or 30.6 mg/kg/day to LE dams GD6-PND21 (oral gavage) (n > 15 per group, n = 8–13 in FOB, n = 7–8 open field, TSH n = 5–12) | Dams PND22: tT4↓ up to ~40%, tT3↔, TSH↑ ~130% in high dose Male pups PD4: tT4↓ 52% in high dose, TSH↔ Male pups PD7: tT4↓ 60% in high dose, TSH↔ Male pups PD14: tT4↓ up to 78%, TSH↔ Male pups PD21: tT4↓ up to 74%, TSH↔ Female pups PD4: tT4↓ 56% in high dose, TSH↔ Female pups PD7: tT4↓ 50% in high dose, TSH↔ Female pups PD14: tT4↓ up to 80%, TSH↔ Female pups PD21: tT4↓ up to 76%, TSH↔ | | FOB including open field PND24, 60 and 273: ↔, except significant interaction dose*age interaction from PD24-60, but no overall treatment effect on PND60 Motor activity in figure eight, PND 24 and 60: ↔ motor activity (motron) PND~110: ↔ | [17] (Kodavanti et al., 2010) |

*(Continued)*

**Table 1.** (Continued)

| Exposure to DE-71 | TH/TSH | Thyroid gland | Developmental neurotoxicity and hearing | Reference |
|---|---|---|---|---|
| 0, 40, 60 mg/kg/day to Wistar dams GD7-PD16 (oral gavage) (n = 19–21) | Dam GD15 tT4↓ up to 60%, tT3↓ up to 25%, TSH↔ Pup PD8-PD27 tT4↓ ~60% all doses, PD16 tT3↓ ~25% all doses, PD27 tT3↓~15% all doses | Dam PD27 thyroid gland weight↔ Female pup PD16 thyroid weight↔ Male pup PD27 thyroid weight↔ Male PD16 histology: no changes in follicular epithelium, follicular morphology, stroma or c-cells but increased minimal vacuolation of follicular colloid in high dose | Motor activity PD21 and PD79: ↔ Morris water maze at 4.5–6 months of age: ↔ Hearing function at 7.5–8 months: ↔ | This study |
| **For comparison: studies with exposure to TPO-inhibitors in same or similar studies** | | | | |
| 0, 0.8, 1.6 or 2.4 mg/kg/day PTU to Wistar dams GD7-PD17 (oral gavage) (n = 18–21) | Dam GD15: tT4↓ with up to ~60% Pup PND16: tT4↓ with up to ~85% | Pups PND16: thyroid gland weight↑ Pup histology PND16: marked changes with hyperplasia and hypertrophy of epithelial cells, papillary projections into lumen and reduced lumen size | Motor activity: PND 14↓, PND17↔, PND23↑, 16 weeks: ↑ Morris water maze, weeks 8–9: ↔ RAM, males 5–6 months: ↓ learning and memory Hearing, adult offspring: ↓ | [10] (Axelstad et al., 2008) |
| 0 or 30 mg/kg/day PTU to Wistar rat pups PND5-22 (oral gavage) (Open field: n = 17, radial maze: n = 11, T4: n = 10) | Male pup PND 23: tT4↓ >95% Female pup PND 23: tT4↓ >95% | PND23: thyroid gland weight↑, both sexes | Brain weight, relative PND23: ↑, both sexes Open field PND42 and 70: ↑motor activity, both sexes RAM PND100: working and reference memory learning deficits, both sexes | [19] (De-Miranda et al., 2016) |
| 0, 8 mg/kg/day MMI, 16 mg/kg/day MMI, 25 or 50 mg/kg/day amitrole SD dams GD7-PD22 (oral gavage) (n = 12). Effects are similar in both MMI and amitrole unless otherwise stated. | Dam GD15: tT4↓ up to ~60%, tT3↓23% in high dose amitrole, TSH↑ up to ~800% Dam PD22: tT4↓ up to ~60% in MMI and 90% in amitrole, tT3↑30% in low dose MMI, TSH↑ up to ~800% Male pup PD16: tT4↓ up to ~80–90% in high dose, tT3↓25–35% in high dose, TSH↑330–425% in high dose Female pup PD17: tT4↓ up to ~80–90% in high dose, tT3↓25–30% in high dose, TSH↑250–360% in high dose | Dam PD22: thyroid gland weight ↑ Male pup PD16: thyroid gland weight↑ Female pup PD17: thyroid gland weight↑ Male pup PD16 histology (n = 4): hyperplasia/hypertrophy, irregular follicles, reduced lumen size and colloid depletion | Motor activity, offspring PND21:↑ | [6] (Ramhøj et al., 2022) |

Abbreviations: f: free, LD: lactation day, LE: Long-Evans, MMI: Methimazole, PND: postnatal day, PD: pup day, PTU: propylthiouracil, RAM: radial arm maze, SD: Sprague-Dawley, T4: thyroxine, T3: 3,3´,5-tri-iodothyronine, TH: thyroid hormones, t: total, TSH: thyroid stimulating hormone

All doses given in mg/kg bodyweight/day.

In Study 2, one male and one female pup per litter were weaned (PD27) and housed pairwise with an animal of the same treatment group (or with a sibling if none were available). These offspring were then used for motor activity assessments on PD79, testing in the Morris water maze for 8 days at 4.5–6 months of age and testing of hearing function on one day at approx. 7–8 months of age, as described below.

Animal experiments were conducted at the DTU National Food Institute's facilities (Mørkhøj, Denmark). Ethical approval was obtained from the Danish Animal Experiments Inspectorate, with authorization number 2012-15-2934-00089 C4. The experiments were overseen by the National Food Institute's in-house Animal Welfare Committee for animal care and use.

## Organ weights, tissue and serum sample collection

In both studies all animals, as specified below, were killed by decapitation under $CO_2/O_2$ anesthesia and trunk blood collected.

Study 1 was terminated on PD14 and livers from the dams excised and weighed. Tongue blood samples were taken from live dams on GD15. Depending on the number of pups in the

litters, blood samples were pooled from 2–4 pups on PD3 and PD8. On PD14, plasma was pooled separately for male and female pups.

In Study 2, remaining pups (except those weaned) and dams were terminated on PD27. From dams liver and thyroid glands were excised and weighed. Necropsies were performed on one male and one female pup per litter on PD16 and PD27. Livers were weighed and on PD16 a piece was fixed in 10% formalin and subsequently processed and embedded in paraffin for histology. Thyroid glands were excised and weighed from PD16 female pups and PD27 male pups. For PD16 males, the thyroid glands were excised attached to a piece of the thyroid cartilage and fixed in 10% formalin, then tissue processed and embedded in paraffin. Tongue blood samples were taken from dams on GD15. Blood was collected from one male and one female pup per litter on PD16. On PD27 blood samples were pooled separately for males and females and included all remaining pups not weaned for later in life tests.

All blood samples were collected in heparinized Eppendorf/vacutainer tubes and stored on ice until centrifugation for 10 min at 4°C and 4000 rpm. The plasma was collected and stored at -80°C until analysis.

## Hormone analysis

Serum total T4 and 3,3′,5-tri-iodothyronine (T3) concentrations were measured at different times in the two studies, as detailed in Fig 1. Total T4 and T3 concentrations were analyzed by electrochemiluminescence-immunoassay (ECLIA)–photoncount using a Cobas 8000 E-modul at the Department of Clinical Biochemistry, Rigshospitalet, Copenhagen, Denmark.

In Study 2 serum TSH were assessed in dam tongue blood on GD15 and in blood from PD16 pups. TSH concentrations were assayed according to manufacturer's instructions using a TSH rat ELISA kit (DEV9977) together with a Rat Control (DEV99RC) both from Demeditec Diagnostics GmbH (Kiel, Germany).

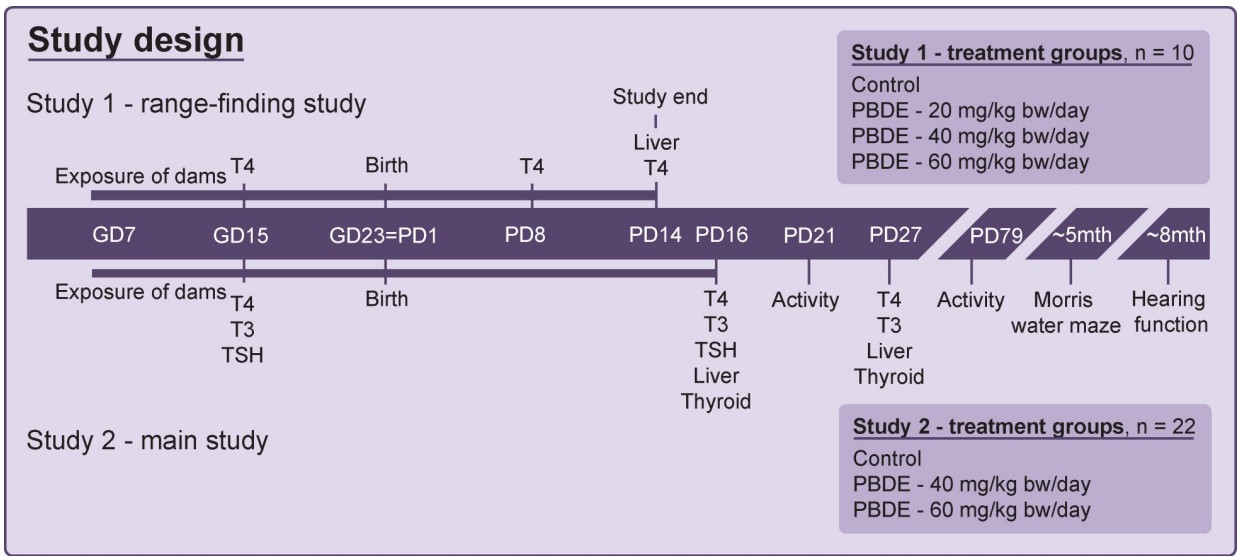

**Fig 1. Study design for two developmental toxicity studies with the DE-71 technical mixture of brominated flame retardants.** Study 1 assessed early postnatal thyroid hormone disruption and toxicity while Study 2 was used for examination of later postnatal thyroid hormone system disruption, liver and later in life effects on behavior, learning and hearing function. Bw: body weight, GD: gestation day, PBDE: polybrominated diphenyl ethers (DE-71), PD: postnatal day. T3: 3,3′,5-tri-iodothyronine, T4: thyroxine. TSH: thyroid stimulating hormone. The toxicity data and reproductive effects are reported elsewhere [25]. Timeline not drawn to scale.

## Histological evaluation of thyroid glands and liver

Two sections each from PD16 thyroid glands and livers were stained with hematoxylin and eosin (H&E) following standard protocols. Thyroid glands of male offspring were assessed for follicle size and shape (presence of papillary projections), vacuolation of follicular colloid, type of follicular epithelium (flattened to columnar and position of nucleus), hyperplasia of follicular epithelium, c-cell hyperplasia, vascularization and fibrosis.

Livers from male and female offspring were assessed for hypertrophy (severity and localization) and vacuolation (severity) of hepatocytes. Vacuolation was defined as large vacuoles displacing the nucleus of hepatocytes.

All histological evaluations were performed by a veterinary pathologist and scoring was done blinded to exposure group.

## Motor activity and habituation

Motor activity and habituation was assessed in prepubertal male and female pup offspring on PD21, and in the weaned adult offspring on PD79. Testing protocols were as previously described [6]. Briefly, each pup was placed in sound and light insulated activity boxes that records horizontal activity via interruptions of adjacent photo beams. Testing took place for 30 min during which the number of photo beam interruptions was registered (via computer in adjoining room) in 10 periods of 3 min. Total counts during the 30 min represents total motor activity levels. Habituation was assessed by the average counts during the initial (0–9 min), middle (10–21 min) and last part of the test (22–30 min). In addition, activity counts were averaged during the first (0–15 min) and second half of the test (16–30 min).

## Learning in the Morris water maze

Learning was assessed in the weaned adult offspring distributed into three new blocks and with testing performed in the Morris water maze with animals ages PD132-PD176 for a period of 5+3 days (separated by two days without testing), in a pool with a diameter of 220 cm. The animals were challenged to learn to find the location of a platform hidden just below the water surface in the pool surrounded by optical cues. The platform was transparent, circular and placed on a solid stand 1 cm below the water surface. Every animal was in four trials per day placed in the pool at four different starting points and allowed to search for the platform. The trial was completed when the animal climbed onto the platform. If the platform had not been located within 60s, the animal was led to the platform by hand and allowed to sit on the platform for 20 seconds before being dried and returned to the cage. Endpoints included latency to locate the platform, path lengths and swimming speeds which where all registered by a camera equipped with a tracking system (Viewpoint video-tracking system, Sandown Scientific, Middlesex, England).

## Auditory function

After the maze testing, a subset of 12 males and females (maximum one of each sex from every litter) per group were subjected to test of hearing function over the course of 4 days between PD216-233. Animals were anesthetized with 1.8 ml/kg bw of a mixture of zolazepam 12.9 mg, tiletamine 12.9 mg, xylazine 1.8 mg, fentanyl 10.3 μg in 0.9% NaCl per 1 ml. Animals roused after approximately 2 hrs and were killed on the same or the following day. Testing procedures and equipment were as previously described [10, 26]. Briefly, hearing was assessed by measuring the distortion product otoacoustic emissions (DPOAE). DPOAE was measured as the amplitude of the cubic distortion product (CDP) from two tones generated by a two-channel

tone generator with phase control (HP 8904) and the probe microphone output feeding into a FFT spectrum analyzer (HP 35670A). The first tone (f1) was always 10 db higher than the second tone (f2) and there was a fixed ratio between them of f2/f1 = 16713 = 1.23. Distortion product diagrams (DP-grams) were generated by measuring CDP at fixed levels of primary tones across frequencies (2 to 70 kHz), based on 64 time-averaged recordings. DPOAE input/ output curves were made at f2 = 4096Hz and 32384 Hz by measuring CDP amplitude at varying levels of primary tones.

## Statistical analysis

Data from continuous endpoints with normal distribution and homogeneity of variance were tested by ANOVA, with appropriate covariates included in the statistical analysis (e.g. body weight for organ weights and motor activity). Dunnett's post-hoc test was applied to account for multiple testing with the statistical significance level set at 0.05. Litter effects were accounted for by only analyzing one pup per litter or by including the litter as an independent random and nested factor in the analysis. Hearing data was analyzed by ANOVA with exposure level and sex as factors, assuming unequal variance between groups.

A 2-sided Fisher's exact test was used for histology data. First a rxk Fisher's exact test was used (including more than 2 groups and more than 2 scores) to detect differences in the distribution of scores in all the groups. To determine if one group differed from controls, a 2xk Fisher's exact test was performed. For dichotomous data, a 2x2 Fisher's test was used to compare two groups.

SAS Enterprise guide 4.3 (2010) (SAS Institute Inc, Cary, NC, USA) and GraphPad Prism 5 (Graphpad Software, San Diego, CA; USA) was used for statistical analysis. Statistical analysis of hearing data was performed in SYSTAT Software Package v. 9.

## Results

### Dam thyroid hormone system disruption

We assayed dam TH system disruption in the pregnant animals. This was done because the fetus is entirely dependent on maternal transfer of T4 to the fetal circulation during early development. On GD15, dam T4 and T3 levels were dose-dependently reduced by DE-71 exposure (<50% and <85% of controls, respectively, for T4 and T3) (Fig 2A and 2B). There were no statistically significant effects on TSH concentrations, albeit values appeared more variable in exposed group compared to controls (Fig 2C).

### Pup thyroid hormone system disruption

Postnatally, brain development depends on the offspring's own TH production. All doses of DE-71 markedly reduced postnatal T4 concentrations to 25–45% of control levels (Fig 3A). On PD16 and PD27, T3 levels were reduced to ~75–85% of controls by DE-71 exposure (Fig 3B). DE-71 exposure was discontinued on PD16, but there was only slow recovery in T4 and T3 levels between PD16 and PD27 (Fig 3A and 3B).

### Effects on the liver

TH system disruption can arise from different mechanisms, one of which is liver enzyme induction (microsomal enzyme inducers). We, therefore, looked for effects in dam and pup livers. Dam liver weights were increased dose-dependently in all exposure groups on PD14 in Study 1 (~20% in high dose) (Fig 4A). The increase was observed for both absolute liver weight

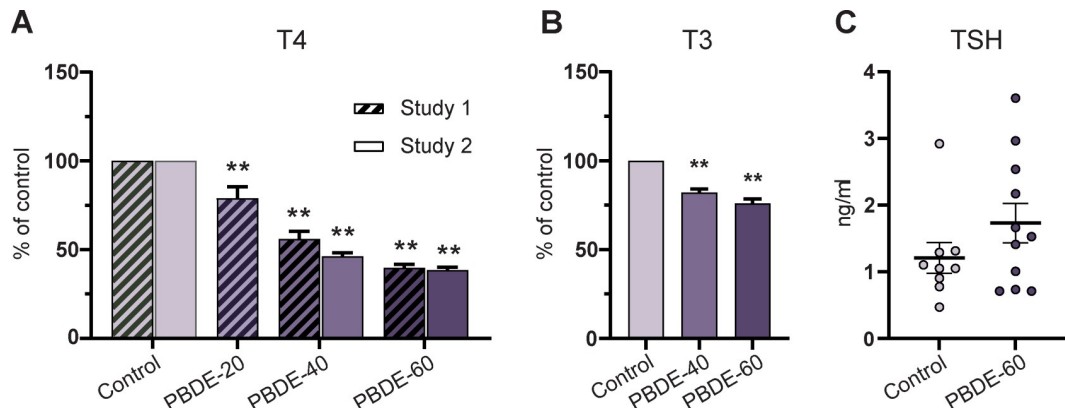

**Fig 2. Dam serum T4, T3 and TSH concentrations on GD15 after 7 days of exposure to DE-71.** (A) Dam serum T4 concentrations as % of control in Study 1 and Study 2. Study 1: n = 8–9 except PBDE-20 with n = 6. Study 2: n = 19–21. Mean + SEM. (B) Dam serum T3 as % of control (Study 2). Mean + SEM. n = 19–21. (C) Dam serum TSH concentrations (Study 2). Mean + SEM. Individual data points with mean and whiskers indicating SEM. n = 9–11. **p<0.01. GD: gestation day, PBDE: polybrominated diphenyl ethers (DE-71). T3: tri-iodothyronine, T4: thyroxine. TSH: thyroid stimulating hormone.

analyzed with body weight as covariate and for relative weights (data in S1 File). There was no effect on body weights. By PD27, in Study 2, liver weights were similar between groups.

Pup liver weights were markedly increased (up to ~50%), particularly on PD16, at the end of exposure (Fig 4B). By PD27 the increase was less pronounced, demonstrating some recovery after exposure ceased. Histological evaluation of pup livers revealed increased hypertrophy and vacuolation of hepatocytes with increasing dose of DE-71 (Fig 4C–4E). Hypertrophy was seen in all exposed animals, and severity increased with increasing dose (Fig 4C). One control male displayed marginal signs of hypertrophy. Hypertrophy of hepatocytes was localised to the centrilobular areas. No microvesicular vacuolation was observed. Vacuolation of hepatocytes was only seen in exposed pups and the severity of vacuolation increased with dose (Fig 4E).

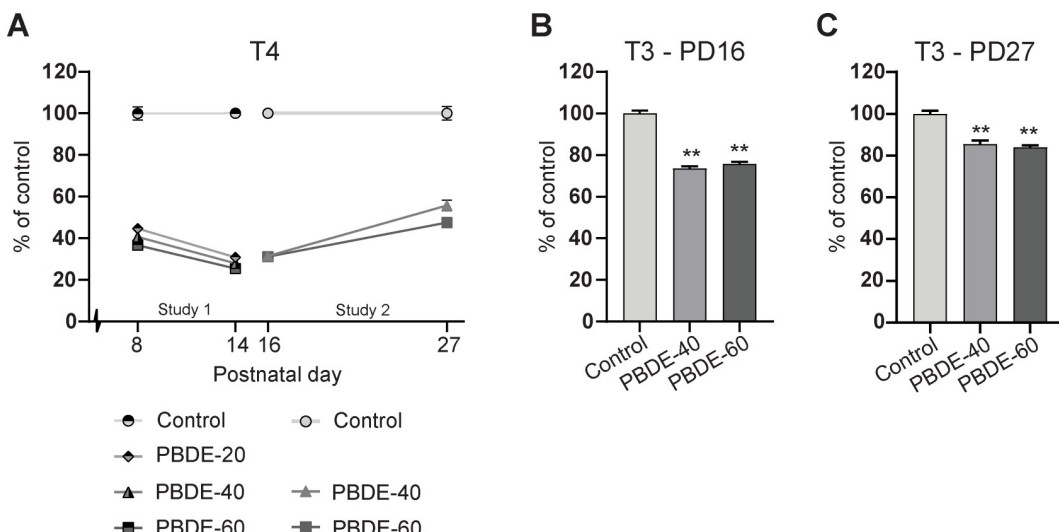

**Fig 3. Pup serum T4 and T3 concentrations during postnatal development.** (A) Pup serum T4 (% of control) in Study 1 and 2 covering PD8 to PD27 T4 was statistically significantly decreased at all doses at all points in time. Study 1: n = 6–8, Study 2: n = 19–21 litters. Mean + SEM. (B) Pup serum T3 concentrations PD16 and PD27 (Study 2). n = 19–21. Mean + SEM. **p<0.01. PBDE: polybrominated diphenyl ethers (DE-71), PD: postnatal day, T3: tri-iodothyronine, T4: thyroxine.

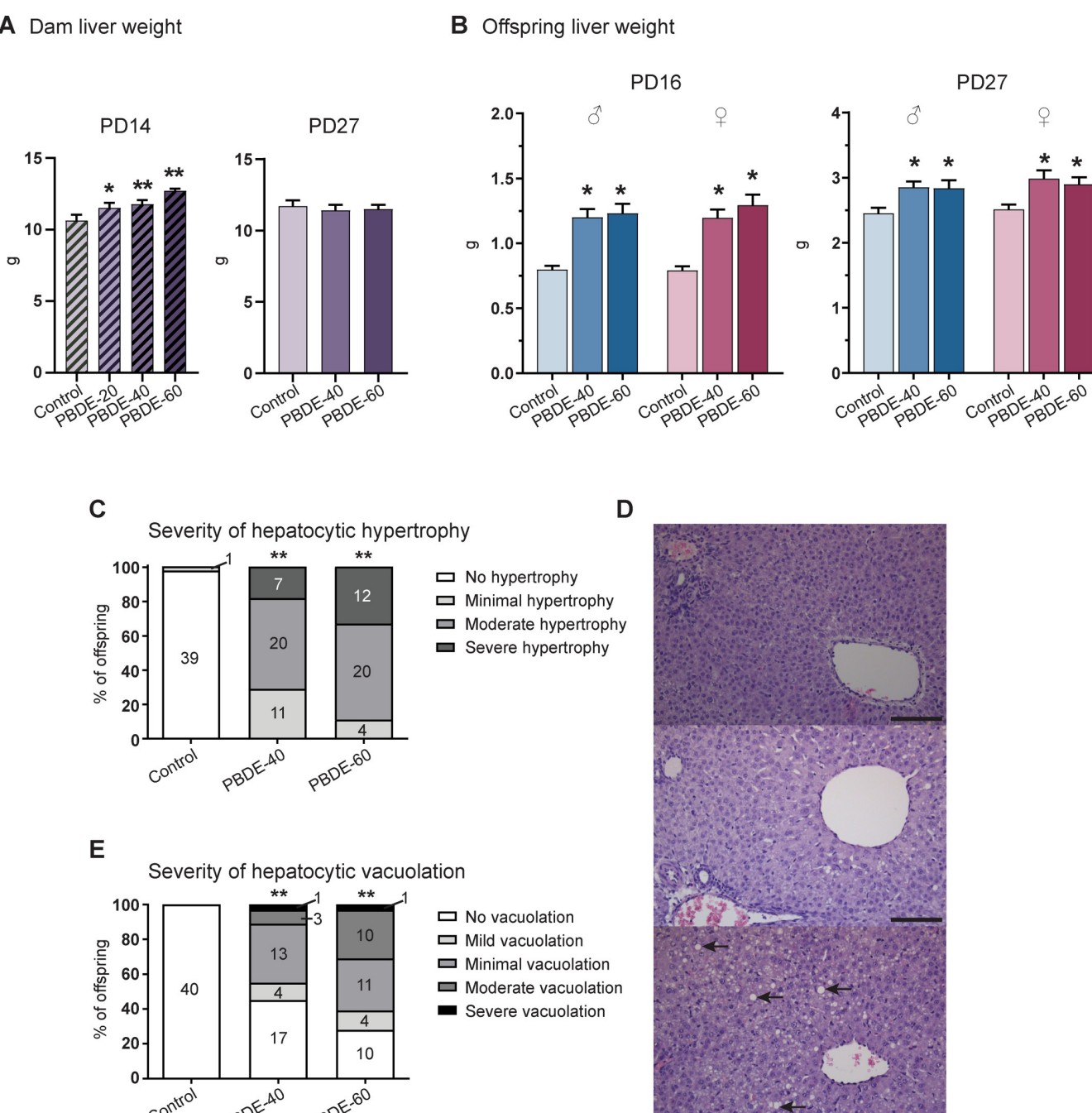

**Fig 4. Effects of DE-71 exposure from GD7-PD14/16 on dam and pup liver weights and histology.** (A) Dam liver weights were increased dose-dependently in Study 1 on PD14, while there were no effects on PD27 in Study 2. n = 7 in Study 1, n = 19–21 in Study 2. Mean + SEM. (B) Male and female pup liver weights PD16 and PD27. n = 17–21. Mean + SEM. (C) Severity of hepatocytic hypertrophy in male and female offspring PD16. Centrilobular hypertrophy was observed in all exposed animals. Number of assessed offspring shown in each bar. n = 36–40. (D) Male PD16 liver from control (top), high-dose presenting with severe centrilobular hepatocytic hypertrophy (middle) and high dose with moderate hepatocytic vacuolation (bottom). Arrows show examples of large macrovesicular vacuoles in hepatocytes (E) Severity of hepatocytic vacuolation in male and female offspring PD16. Significantly increased incidence of vacuolation was seen in exposed offspring. Number of assessed offspring shown in each bar. n = 36–40. *p<0.05, **p<0.01. Scalebar = 100 μm. PBDE: polybrominated diphenyl ethers (DE-71), PD: postnatal day.

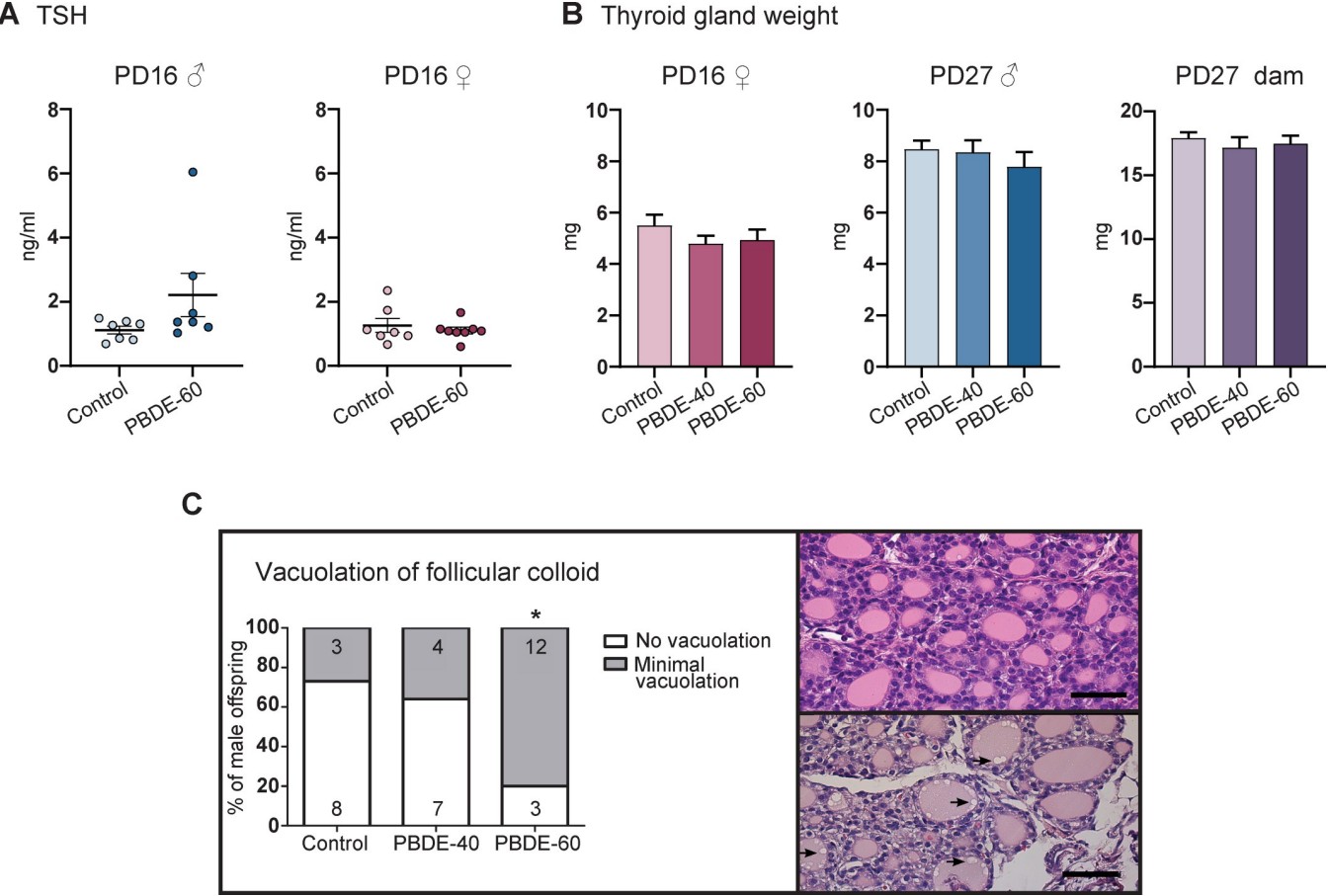

**Fig 5. Effects on the hypothalamic-pituitary-thyroid axis in rat offspring exposed to DE-71 during development.** (A) TSH concentrations in PD16 male and female offspring. Individual data points shown with mean and SEM whiskers. n = 7–8. (B) Thyroid gland weights in female offspring PD16, male PD27 and dams PD27. Females: n = 19–21, males: n = 15–17, Dams: n = 19–21. (C) Vacuolation of thyroid gland follicular colloid in male PD16 offspring. Left: incidences of vacuolation and severity, number of assessed male pups shown inside bars. Upper right: Thyroid gland with no vacuolation of follicular colloid. Lower right: thyroid gland with minimal vacuolation (arrows) of follicular colloid. *p< 0.05. Scalebar = 50 μm. PBDE: polybrominated diphenyl ethers (DE-71), PD: postnatal day, TSH: thyroid-stimulating-hormone.

## The hypothalamic-pituitary-thyroid axis

TH system disruption by microsomal enzyme inducers appears to cause two different effect patterns within the HPT-axis. Some enzyme inducers, such as phenobarbital, reduce serum T4 concentrations with a concurrent activation of the HPT-axis, resulting in increased TSH levels to stimulate TH production in the thyroid gland [27, 28]. This leads to increased thyroid gland weight and histological changes. In this study, DE-71 exposure reduced T4 without affecting TSH levels or thyroid gland weights, either in PD16 and PD27 pups (Fig 5A and 5B) or PD27 dams (Figs 2C & 5B). No changes in the follicular epithelium, follicular morphology, stroma or c-cells were evident in thyroid glands of PD16 male pups. However, we observed a dose-dependent increase in vacuolation of follicular colloid with a statistically significant higher incidence in the high dose-group compared to controls (Fig 5C). The vacuolation was negligible and no vacuolation of the follicular epithelium was observed in relation to vacuolation of colloid. This could be signs of increased activation of the thyroid follicles. However, the lack of effects on epithelium and follicular morphology indicates that the activation was minor, if any.

Taken together, our results indicate that the DE-71 mixture is a microsomal enzyme inducer which does not cause an activation of the HPT-axis in rat offspring.

### Offspring neurobehavior

Motor activity levels and habituation were tested in prepubertal (PD21) and adult (PD79) off-spring. We found habituation (decreasing activity levels over the course of the test) at both ages in males and females and observed the expected sex difference in adulthood, i.e. higher activity levels in females than in males. There were no significant treatment-related effects of developmental DE-71 exposure on general activity (Fig 6), nor on habituation in males (Fig 6) or females (data in S1 File) at any age. Learning was tested in the Morris water maze at approximately 5 months of age. We found improved performance over the course of the test, demonstrating a functional assay. However, we found no effects of developmental exposure on distance, latency to reach platform or swimming speed (Fig 6C). This indicates that DE-71 exposure did not affect learning as assessed by this simple behavioral assay.

### Offspring hearing function

We have previously shown how developmental hypothyroidism results in increased hearing thresholds and decreased Cubic Distortion Product (CDP) with loss of activity in the outer hair cells spread equally over the basilar membrane in the cochlea [10]. We performed the same tests on the DE-71 exposed animals but found no differences compared to control animals (Fig 7).

## Discussion

One of the main aims of this study was to clarify if there is a dose dependent correlation between low serum T4 levels caused by developmental exposure to DE-71 and neurobehavioral changes in the rat offspring. Based on our results, and those of previous studies, the answer appears to be no. This is despite the fact that perinatal exposure to DE-71 markedly reduces plasma T4 and T3 concentrations throughout development. Notably, however, the HPT-axis does not seem to be activated by this marked reduction in T4 as, for instance, is the case with hypothyroidism induced by propylthiouracil (PTU) [10]. In PTU-exposed animals, TSH levels increase in response to a drop in T4. Regardless, a severe reduction in T4 did not result in adverse effects on behavior or hearing. It thus appears that thyroid hormone system disruption by DE-71 does not result in the severe hypothyroidism and associated effects that can be ascribed to PTU and other potent TPO-inhibiting compounds [6–8, 10].

As discussed in the introduction, several studies have examined the potential effects of DE-71 on the thyroid hormone system and neuroendocrine development. However, it is challenging to draw general conclusions since study designs vary considerably. This is also the conclusion reached by a systematic review of developmental neurotoxicity by PBDEs in animal studies [29]; nevertheless the evidence for neurotoxicity in humans is quite strong [30–33]. In Table 1, we have summarized the key rat studies of DE-71 and reported effects in order to integrate our new results in a more holistic analysis. From this synthesis it is evident that DE-71 induces very few effects on behavioral endpoints across developmental toxicity studies.

There is also an obvious decoupling between low serum T4 levels and changes to brain development as assessed in standard neurobehavioral tests. The dose-dependent correlations seen with TPO inhibitors simply do not occur with DE-71 exposure.

One of the primary target organs of PBDEs is the liver. We found a pronounced increase in liver weights and histological changes in DE-71 exposed animals, which is consistent with microsomal enzyme induction reported by others. More specifically, DE-71 can induce

## A  Prepubertal offspring

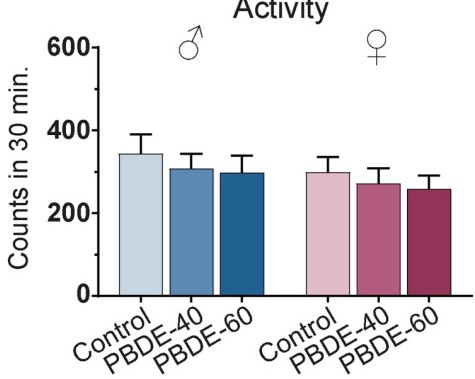

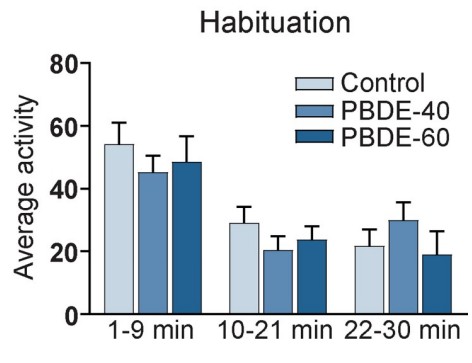

## B  Adult offspring

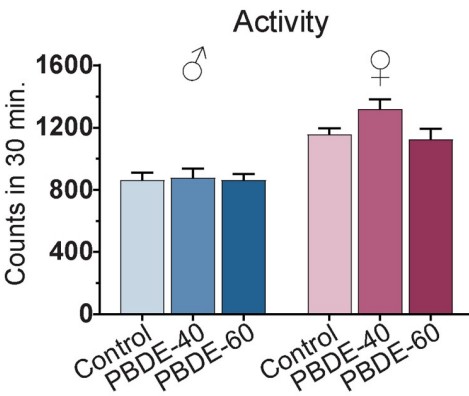

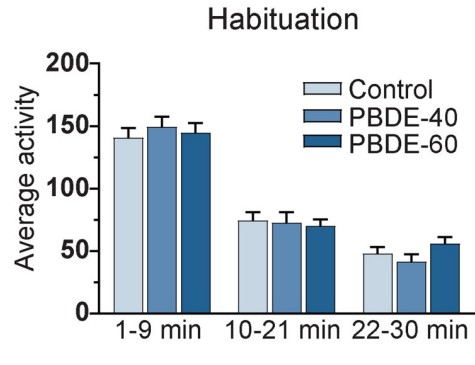

## C  Morris water maze

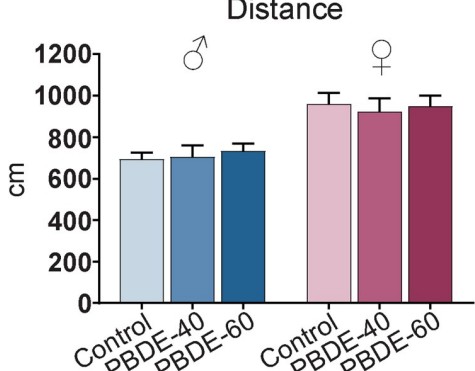

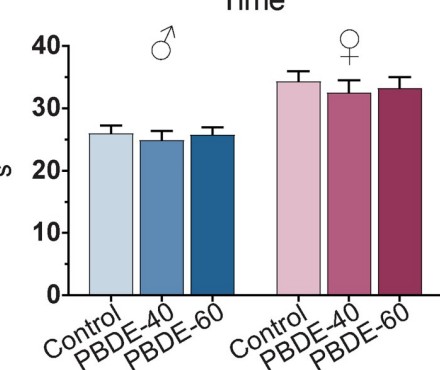

**Fig 6. Neurobehavior in rat pups exposed to PBDEs during development.** (A) Total motor activity (total counts in 30 min) levels in PD21 male and female offspring and habituation (average activity counts per 3 min period) in male offspring. Control and PBDE-40: n = 18–19, PBDE-60: n = 14–15. (B) Total motor activity levels in PD79 adult male and female offspring and habituation in male offspring. Control and PBDE-40: n = 17–20, PBDE-60: n = 15. (C) Mean distance travelled and time spent to reach hidden platform in the Morris water maze (8 days of testing at 4.5–6 months of age). n = 18–20. Mean + SEM. PBDE: polybrominated diphenyl ethers (DE-71), PD: postnatal day.

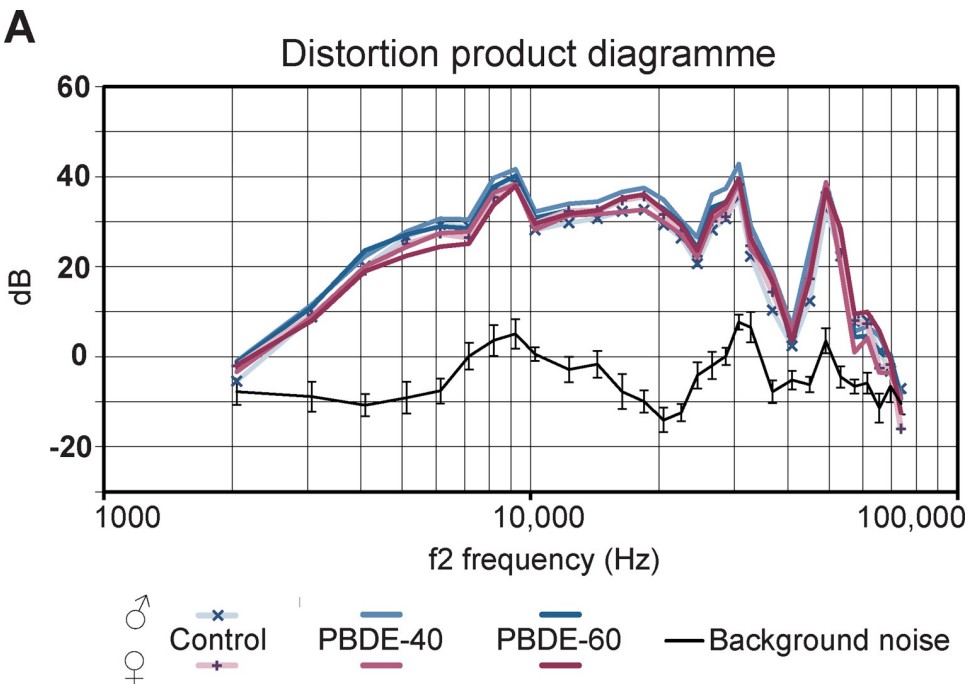

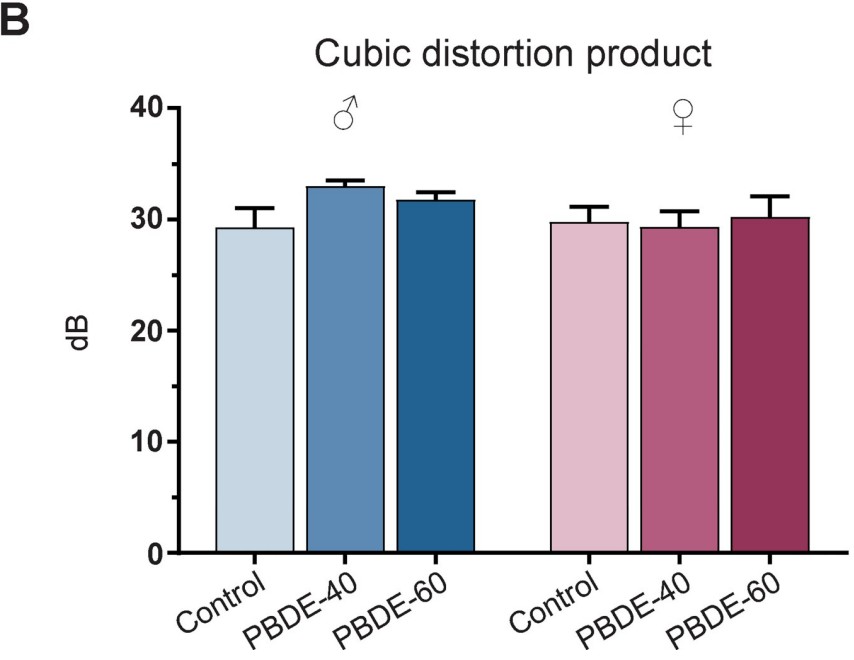

**Fig 7. Hearing function in adult 7.5–8 months old male and female offspring exposed to PBDEs during development.** (A) Distortion product diagram of cubic distortion product at all tested frequencies (2 to 70 kHz). Data represent group means, black curve represents background noise ± 95% confidence interval. (B) Average cubic distortion product for f2 at 4 and 32 kHz. PBDE: polybrominated diphenyl ethers (DE-71). n = 11–12.

enzymes related to TH metabolism and excretion [4, 35–37]. This suggests that increased hepatic clearance of THs contributes to the reduction in serum T4. Notably, PBDEs also possess the ability to bind and displace T4 from the serum thyroid hormone transport protein

TTR, with hydroxylated metabolites of PBDEs having particularly high affinity [38–40]. *In vivo* these two mechanisms may act together to increase excretion of thyroid hormones and produce the phenotype of low serum T4 concentrations. How this relates to the lack of HPT-axis activation, however, remains unexplained.

DE-71 exposure severely reduced serum T4 concentration throughout postnatal development. Comparing this to other studies with developmental exposure to DE-71 (Table 1) shows that there is concordance between studies and strong evidence to conclude that DE-71 consistently reduces serum T4 in perinatally exposed pups. It also correlates with T4 effects in similar developmental toxicity studies with TPO inhibitors. With regard to T3, however, few studies have found, or even measured, any effect on serum concentrations in pups exposed to DE-71. Three studies with small group sizes report no effect on T3 levels [4, 24, 35], while one report reduced serum T3 in both male and female pups on PD21 [18] (Table 1). We also found reduced serum T3 levels in PD16 pups measured across relatively large group sizes of 19–21. This strengthens the evidence for DE-71 causing a reduction in pup T3 concentrations, yet the available data is inconsistent.

The thyroid gland is the site of TH synthesis and the target of TSH and thus natural to assess for potential thyroid hormone system disruption and HPT-axis activation. That is, thyroid gland weight and histology should be scrutinized even though DE-71 does not appear to increase TSH levels in the majority of studies. With regard to TSH, our study, together with two others, observed no effect on pup serum TSH [17, 24] while a fourth study reported a 30% increase [18]. This latter study also found follicular epithelium cell height to be increased in PD21 pups, but did not investigate other adverse effects in the thyroid glands [18] (Table 1). In fact, the only other assessment of thyroid glands is a study that found no effects on pup thyroid gland weight after DE-71 exposure [19] (Table 1). Thus, assessments of thyroid gland weight and histology are scarce and our study is the first to report on a full histological assessment of DE-71-exposed pup thyroid glands. This assessment revealed no marked effects, which is in contrast to what is reported in PD16 pups developmentally exposed to PTU, MMI or amitrole [6, 10, 41, 42]; however, we did find minimal effects on colloid vesicles. Taken together, exposure to DE-71 does not appear to activate the HPT-axis in perinatally exposed rat pups, and only induces marginal, if any, changes to thyroid gland morphology.

Growing evidence suggests that rodent models for hypothyroxinemia-like effect patterns are not sufficiently sensitive to determine if a TH system disrupting chemical possess developmental neurotoxicity potential [1]. That is, the effects on the brain, if present, are not pronounced enough to be detected by standard neurotoxicity assays, but the TH system disrupting potential is great enough to be of concern to human health. The fact is that our current experimental conditions limit our ability to measure and determine if there is an effect or not due to endpoint sensitivity. Firstly, dose-response studies of effects of TH deficiency induced by TPO inhibitors show that serum and brain THs need to be severely reduced in foetuses and postnatal pups before statistically significant effects on brain development are observed [1, 6, 9, 10, 12, 42, 43]. Thus, current endpoints for neurotoxicity are not sensitive enough to detect developmental TH system disruption. Secondly, there likely is a discrepancy between serum and brain TH concentrations in rodents subject to chemically induced hypothyroxinemia-like conditions. Here, brain hormones can be less affected despite pronounced effects on serum hormones [14], opposite to the relationships seen with PTU exposure [9, 44]. Whether this also applies to DE-71 exposed animals should be investigated in future studies. Regardless, the simple neurobehavioral tests used herein, and in other studies, cannot identify adverse neurological effects of DE-71 induced TH-system disruption in rat offspring.

While our neurobehavioral assays, and potentially also the rodent brain, may be relatively insensitive to TH deficiency, there is no question that the situation is very different for the human brain. In humans, severe lack of TH during development has profound consequences for brain development. All the while there is also a substantial body of literature showing that even slightly reduced serum T4 with no effects on TSH, so-called hypothyroxinemia, is enough to impair child brain development. This is evident from epidemiological studies, that find decreased IQ, altered white-to-grey matter ratio, altered motor and language development, and increased risk of developing neurobehavioral disorders such as autism, ADHD and schizophrenia [45–51]. Thus, while it is challenging to test rodent brain function for effects of TH system disruption, the evidence from humans is that hypothyroxinemia does negatively impact human brain development. Accordingly, the widespread exposure of humans to TH system disrupting chemicals is of great concern if we are to protect human cognition and health for future generations.

## Concluding remarks

Perinatal exposure to DE-71 reduces serum T4 markedly in postnatal rat pups, but without a concomitant increase in serum TSH, i.e. a hypothyroxinemia-like response. Based on data herein and from historical studies, there is no clear correlation between significantly lower serum T4 levels and adverse effects as assessed by standard neurobehavioral tests in rats. This contrasts with the correlations seen for TH system disruption by TPO inhibitors, where low serum T4 correlates with TH-mediated effects on the brain. To address this conundrum, future studies should aim to measure TH levels also in target tissues, such as the brain, and include endpoints that are specific to TH system disruption. Only then can we answer the outstanding question if the absence of adverse neurobehavioral effects in DE-71 exposed pups are due to decoupling of systemic TH concentrations and local TH concentrations in the brain. Importantly, however, it should be noted that a reduction in serum THs is in itself of great concern to human health. From what we know, the human brain is most likely very sensitive to any alteration in thyroid status of pregnant women and their developing foetuses. In humans TH system disruption of any kind–hypothyroidism, hypothyroxinemia, or otherwise–is cause for concern as regards brain development.

## Supporting information

**S1 File. Datasets.**
(XLSX)

## Acknowledgments

We would like to thank Tommy Licht Cederberg (National Food Institute, Technical University of Denmark) for the DE-71 congener analysis.

We thank Mette Voigt Jessen, Heidi Broksø Letting, Lillian Sztuk, Dorte Lykkegaard Korsbech, Birgitte Møller Plesning, Ulla El-Baroudy, Sarah Grundt Simonsen, Lene Ravn, Eva Terrida and Lasse Laub-Ekgreen for excellent technical assistance. We also thank Anne Ørngreen and staff for animal care and husbandry.

## Author Contributions

**Conceptualization:** Louise Ramhøj, Ulla Hass, Marta Axelstad.

**Formal analysis:** Louise Ramhøj, Karen Mandrup, Søren Peter Lund, Karin Sørig Hougaard, Marta Axelstad.

**Funding acquisition:** Ulla Hass, Marta Axelstad.

**Investigation:** Louise Ramhøj, Karen Mandrup, Søren Peter Lund, Marta Axelstad.

**Visualization:** Louise Ramhøj, Karen Mandrup.

**Writing – original draft:** Louise Ramhøj, Terje Svingen, Marta Axelstad.

**Writing – review & editing:** Louise Ramhøj, Terje Svingen, Karen Mandrup, Ulla Hass, Søren Peter Lund, Anne Marie Vinggaard, Karin Sørig Hougaard, Marta Axelstad.

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
