## [Decision Letter · Decision Letter 0]

1 Jun 2022

PONE-D-22-07082Developmental exposure to the brominated flame retardant DE-71 reduces serum thyroid hormones in rats without hypothalamic-pituitary-thyroid axis activation or neurobehavioral changes in offspringPLOS ONE

Dear Dr. Ramhøj,

Thank you for submitting your manuscript to PLOS ONE. After careful consideration, we feel that it has merit but does not fully meet PLOS ONE’s publication criteria as it currently stands. Therefore, we invite you to submit a revised version of the manuscript that addresses the points raised during the review process.

We look forward to receiving your revised manuscript.

Kind regards,

Jamie C. DeWitt

Academic Editor

PLOS ONE

Journal Requirements:

“This study was funded by the Danish Environmental Protection Agency, Ministry of Environment and Food of Denmark. KS Hougaard received support from FFIKA, Focused Research Effort on Chemicals in the Working Environment, from the Danish Government. The funders had no role in study design, data collection and analysis, decision to publish, or preparation of the manuscript.”

Please note that funding information should not appear in the Funding section or other areas of your manuscript. We will only publish funding information present in the Funding Statement section of the online submission form.

“This study was funded by the Danish Environmental Protection Agency, Ministry of Environment and Food of Denmark. KS Hougaard received support from FFIKA, Focused Research Effort on Chemicals in the Working Environment, from the Danish Government. The funders had no role in study design, data collection and analysis, decision to publish, or preparation of the manuscript.”

4.  We note that you have included the phrase “data not shown” in your manuscript. Unfortunately, this does not meet our data sharing requirements. PLOS does not permit references to inaccessible data. We require that authors provide all relevant data within the paper, Supporting Information files, or in an acceptable, public repository. Please add a citation to support this phrase or upload the data that corresponds with these findings to a stable repository (such as Figshare or Dryad) and provide and URLs, DOIs, or accession numbers that may be used to access these data. Or, if the data are not a core part of the research being

5. Please ensure that you refer to Figure 6 in your text as, if accepted, production will need this reference to link the reader to the figure

6. Please include captions for your Supporting Information files at the end of your manuscript, and update any in-text citations to match accordingly. Please see our Supporting Information guidelines for more information: http://journals.plos.org/plosone/s/supporting-information

Additional Editor Comments:

Reviewers have made several suggested modifications to the submitted manuscript for clarity and completeness. Please be sure to respond to all reviewer recommendations.

Reviewers' comments:

Reviewer's Responses to Questions

**Comments to the Author**

1. Is the manuscript technically sound, and do the data support the conclusions?

Reviewer #1: Yes

Reviewer #2: Yes

2. Has the statistical analysis been performed appropriately and rigorously? 

Reviewer #1: Yes

Reviewer #2: Yes

3. Have the authors made all data underlying the findings in their manuscript fully available?

Reviewer #1: Yes

Reviewer #2: Yes

4. Is the manuscript presented in an intelligible fashion and written in standard English?

Reviewer #1: Yes

Reviewer #2: Yes

5. Review Comments to the Author

Reviewer #1: Minor editorial revisions will be needed.

1. Line 116: add St. Louis, MO, USA after Aldrich

2. Line 148: add company name, city, state, country after Tecniplast

3. Line 302: add comma before and after therefore

4. Lines 530-531, 542-543, 557-558, 574-575, 601-602, 615-617, 619-620, 653-654, 682, 686-687: titles should be small capitals

5. Line 535: Vitro should be vitro

6. Line 543 should be fixed appropriately

7. Line 568-569: Journal name should be appropriately abbreviated

8. Line 624: Heal should be Health

9. Line 637: pbfe should be PBDE

Reviewer #2: 1. Abstract, overall. The Abstract could be rewritten to use more precise language. As currently written, many parts are very general. For example, lines 38-40 do not indicate if this effect was in dams or offspring. Similarly, lines 40-41 indicate that no effects in brain function were observed by standard behavioral assays, but brain function is challenging to evaluate with just behavioral assays. Therefore, it is recommended that the Abstract be rewritten for precision.

2. Introduction (and in Discussion as well). It is suggested that the authors be more precise in the Introduction with respect to use of the term “marked.” It would improve the Introduction if the authors could include effect sizes/percent differences reported for these changes in levels of T4. By how much DE-71 reduces T4 would help readers understand the biological significance of this reported effect. This is done in line 84 and would be helpful if done throughout the Introduction.

3. Introduction, overall. It is recommended that the authors include a short paragraph in the Introduction about PBDEs in general and about DE-71 specifically. Why should this class of chemicals be studied for its developmental effects? What is its use? What is its environmental fate and transport? How are people exposed? What are human health concerns? This additional context would enhance the Introduction.

4. Methods, general. In the Introduction the authors indicate that one reason for undertaking the studies described in the manuscript was to determine if DE-71 affected the developing TH system similarly to TPO-inhibitors. However, it does not appear as if a TPO-inhibitor positive control was included as an experimental group. It is recommended that the authors provide data from such a group if it exists or include a brief statement of rationale about why such a group is not a necessary part of the experimental design.

5. Methods, line 150. Please indicate at what age were the one male and one female pup/litter weaned for Study 2.

6. Methods, lines 167-169. Please describe blood collections from pups on PD3 and PD8.

7. Methods, histopathology. Please provide more information on histopathological examinations, including whether a veterinary pathologist did the scoring, whether evaluators were blinded to dose groups, etc.

8. Methods, behavioral testing. It would be helpful if authors could include a brief statement of rationale for the behavioral tests chosen for their relevance to assessing impacts of TH on brain development.

9. Results, throughout. It is recommended that the authors provide context for terms such as “reduced,” i.e., by including percent differences or effect sizes.

10. Discussion, Table 1. This will make a nice addition to the literature on developmental neurotoxicity of PBDE.

11. Discussion, overall. The Discussion is very nicely put together.

6. PLOS authors have the option to publish the peer review history of their article (what does this mean?). If published, this will include your full peer review and any attached files.

Reviewer #1: **Yes: **Taisen Iguchi

Reviewer #2: No

---

## [Author Response · Author response to Decision Letter 0]

13 Jun 2022

Response to Reviewer’s Comments - PONE-D-22-07082

We thank the reviewers for their assessment of our manuscript and their insightful suggestions. We have responded to all comments below and amended the manuscript accordingly. 

Reviewers' comments:

Academic editor: Reviewers have made several suggested modifications to the submitted manuscript for clarity and completeness. Please be sure to respond to all reviewer recommendations.

Thank you, we have revised the manuscript and believe that we have addressed all comments as detailed below and in the revised manuscript.

Reviewer #1: Minor editorial revisions will be needed.

Thank you for pointing out these errors and omissions. They are all now corrected in the revised manuscript.

1. Line 116: add St. Louis, MO, USA after Aldrich

2. Line 148: add company name, city, state, country after Tecniplast

3. Line 302: add comma before and after therefore

4. Lines 530-531, 542-543, 557-558, 574-575, 601-602, 615-617, 619-620, 653-654, 682, 686-687: titles should be small capitals

5. Line 535: Vitro should be vitro

6. Line 543 should be fixed appropriately

 The citation has been updated and it follows PLOS ONE citation for websites.

7. Line 568-569: Journal name should be appropriately abbreviated

8. Line 624: Heal should be Health

Health is abbreviated to Heal in the PLOS ONE citation style.

9. Line 637: pbfe should be PBDE

Reviewer #2: 

1. Abstract, overall. The Abstract could be rewritten to use more precise language. As currently written, many parts are very general. For example, lines 38-40 do not indicate if this effect was in dams or offspring. Similarly, lines 40-41 indicate that no effects in brain function were observed by standard behavioral assays, but brain function is challenging to evaluate with just behavioral assays. Therefore, it is recommended that the Abstract be rewritten for precision.

A1: We have revised the abstract to be more specific including in which animals effects were observed. We have also specified that our study concerns standard behavioral assays and, as the reviewer, we conclude that we lack more sensitive assays: “thus we propose that we lack assays to identify developmental neurotoxicity caused by chemicals disrupting the TH system through various mechanisms”. 

2. Introduction (and in Discussion as well). It is suggested that the authors be more precise in the Introduction with respect to use of the term “marked.” It would improve the Introduction if the authors could include effect sizes/percent differences reported for these changes in levels of T4. By how much DE-71 reduces T4 would help readers understand the biological significance of this reported effect. This is done in line 84 and would be helpful if done throughout the Introduction.

A2: Thank you for this suggestion, we have specified the language and % reductions in the introduction.

3. Introduction, overall. It is recommended that the authors include a short paragraph in the Introduction about PBDEs in general and about DE-71 specifically. Why should this class of chemicals be studied for its developmental effects? What is its use? What is its environmental fate and transport? How are people exposed? What are human health concerns? This additional context would enhance the Introduction.

A3: We have added a paragraph in the introduction on this issue (lines 73-79): “Now banned, PBDEs were for decades used as flame retardants in both industrial and consumer products. Today, human exposure continues due to environmental contamination, persistence and bioaccumulation. Thus, the primary exposure routes are food and house dust [20]. DE-71 is a technical mixture of PBDEs and its congeners are still found in human serum, breast milk and house dust [20,21]. In humans and animals, exposure to PBDEs has been associated with a range of effects including neurotoxicity, disruption of the reproductive and thyroid hormone systems”

4. Methods, general. In the Introduction the authors indicate that one reason for undertaking the studies described in the manuscript was to determine if DE-71 affected the developing TH system similarly to TPO-inhibitors. However, it does not appear as if a TPO-inhibitor positive control was included as an experimental group. It is recommended that the authors provide data from such a group if it exists or include a brief statement of rationale about why such a group is not a necessary part of the experimental design.

A4: Thank you, we agree that it was a bit unclear and have added a sentence in the introduction line 105-107 to indicate that TPO-inhibitor induced effects serve as positive control: “The study design and endpoints are aligned with our previous studies of TPO inhibitors propylthiouracil, methimazole and amitrole [6,10] (refer to Table 1 for summary results). Also, the discussion elaborates on this issue.

5. Methods, line 150. Please indicate at what age were the one male and one female pup/litter weaned for Study 2.

A5: specified to PD27.

6. Methods, lines 167-169. Please describe blood collections from pups on PD3 and PD8.

A6: thank you, lines 174-175 specify that all blood samples are trunk blood. The following description pertains to ages, pooling and sex. We realize that the statement lines 165-166 were unclear and have rewritten the text so that it is now clearly stated that all blood collections are trunk blood. “In both studies all animals, as specified below, were killed by decapitation under CO2/O2 anesthesia and trunk blood collected”.

7. Methods, histopathology. Please provide more information on histopathological examinations, including whether a veterinary pathologist did the scoring, whether evaluators were blinded to dose groups, etc.

A7: On lines 215-217 we have added “All histological evaluations were performed by a veterinary pathologist and scoring was done blinded to exposure group.”

8. Methods, behavioral testing. It would be helpful if authors could include a brief statement of rationale for the behavioral tests chosen for their relevance to assessing impacts of TH on brain development.

A8: see A4

9. Results, throughout. It is recommended that the authors provide context for terms such as “reduced,” i.e., by including percent differences or effect sizes.

A9: Specification of effects sizes have been added to the results on pages 12-13.

10. Discussion, Table 1. This will make a nice addition to the literature on developmental neurotoxicity of PBDE.

A10: Thank you for this assessment.

11. Discussion, overall. The Discussion is very nicely put together.

A11: We thank the reviewer for the thoughtful comment.

---

## [Editor Report · Decision Letter 1]

4 Jul 2022

Developmental exposure to the brominated flame retardant DE-71 reduces serum thyroid hormones in rats without hypothalamic-pituitary-thyroid axis activation or neurobehavioral changes in offspring

PONE-D-22-07082R1

Dear Dr. Ramhøj,

We’re pleased to inform you that your manuscript has been judged scientifically suitable for publication and will be formally accepted for publication once it meets all outstanding technical requirements.

Kind regards,

Jamie C. DeWitt

Academic Editor

PLOS ONE

Additional Editor Comments (optional):

Thank you for thoughtfully responding to reviewer concerns.
---

## [Editor Report · Acceptance letter]

7 Jul 2022

PONE-D-22-07082R1 

Developmental exposure to the brominated flame retardant DE-71 reduces serum thyroid hormones in rats without hypothalamic-pituitary-thyroid axis activation or neurobehavioral changes in offspring 

Dear Dr. Ramhøj:

I'm pleased to inform you that your manuscript has been deemed suitable for publication in PLOS ONE. Congratulations! Your manuscript is now with our production department. 

Kind regards, 

on behalf of

Dr. Jamie C. DeWitt 

Academic Editor

PLOS ONE